# Intelligent Sensors for dc Fault Location Scheme Based on Optimized Intelligent Architecture for HVdc Systems

**DOI:** 10.3390/s22249936

**Published:** 2022-12-16

**Authors:** Muhammad Zain Yousaf, Muhammad Faizan Tahir, Ali Raza, Muhammad Ahmad Khan, Fazal Badshah

**Affiliations:** 1School of Electrical and Information Engineering, Hubei University of Automotive Technology, Shiyan 442002, China; 2School of Electric Power, South China University of Technology, Guangzhou 510630, China; 3School of Electrical Engineering, University of Engineering and Technology, Lahore 39161, Pakistan; 4School of Electrical and Information Engineering, Tianjin University, Tianjin 300072, China

**Keywords:** Levenberg–Marquardt backpropagation, protection sensor, Bayesian optimization, modular multilevel converter

## Abstract

We develop a probabilistic model for determining the location of dc-link faults in MT-HVdc networks using discrete wavelet transforms (DWTs), Bayesian optimization, and multilayer artificial neural networks (ANNs) based on local information. Likewise, feedforward neural networks (FFNNs) are trained using the Levenberg–Marquardt backpropagation (LMBP) method, which multi-stage BO optimizes for efficiency. During training, the feature vectors at the sending terminal of the dc link are selected based on the norm values of the observed waveforms at various frequency bands. The multilayer ANN is trained using a comprehensive set of offline data that takes the denoising scheme into account. This choice not only helps to reduce the computational load but also provides better accuracy. An overall percentage error of 0.5144% is observed for the proposed algorithm when tested against fault resistances ranging from 10 to 485 Ω. The simulation results show that the proposed method can accurately estimate the fault site to a precision of 485 Ω and is more robust.

## 1. Introduction

To date, China celebrates the completion of 30,000 km of ultra-high-voltage lines connecting six regional grids with a total transmission capacity of close to 150 gigawatts [1]. However, power engineers struggle to manage and regulate the impact of dc-link faults in hybrid ac/dc systems [2]. Let us say the 8 GW dc-link from Gansu reports a fault unexpectedly, and the protection algorithm cannot locate it. The power outage might start a chain reaction, resulting in widespread blackouts throughout Hunan and beyond. As a result, ensuring accurate fault location is beneficial to minimize the threat of possible failure and is a prerequisite for the successful and safe operation of dc transmission systems [3]. Furthermore, accurate fault location estimation is important for maintaining the voltage stability of the power system [4] and operating the electricity market efficiently [5].

The prediction of correct fault sites in dc transmission systems has been shown to be reliable by frequency extraction, fault signal analysis, and travelling-wave (TW) approaches in previous studies [2,3,6]. Currently, TW methods based on the concept of travelling-wave reflections are preferred in dc transmission projects since they are highly accurate, reliable, and have high fault resistance [7]. The advancement of TW theory has led to the development of several signal processing techniques, such as wavelet transformation (WT) [8], S and Hilbert–Huang transform [6,9], empirical mode decomposition (EMD) [10], etc. A waveform’s characteristics are analyzed using approximate or detailed coefficients in WT to predict fault locations. However, conventional TW methods require a very high sampling frequency to accurately predict fault location, which leads to expensive computation in the power grid [11].

Researchers have begun exploring intelligent algorithms to rectify computation burden and noise handling capabilities [12]. With the alienation coefficient and the Wigner distribution function [13], an effective transmission line protection mechanism for underground cables is proposed in [14] and implemented for a renewable-energy-based grid. In addition, machine-learning tools such as the radial basis function neural network (RBFNN) [15], support vector machine (SVM) [16,17], extreme machine learning [18], k-means cluster [19], etc., are also utilized. These algorithms can self-learn and modify weights and thresholds while training with historical data. Therefore, they are suitable for complex networks such as multiterminal high-voltage direct-current (MMC-MT-HVdc) systems, where constructing a feasible intelligent algorithm can be challenging.

Because of the enlargement of the structure and the extraordinary growth in the number of learning parameters in MMC-MT-HVdc networks, these intelligent algorithms experience slow convergence and computational burden [20]. Downsampling learning parameters may resolve the above issue but may eliminate valuable features. It is, therefore, imperative to find an algorithm for fault location with high accuracy, minimal computational burden, and low sensitivity to noise [21,22]. With this in mind, this work combines the discrete wavelet transform (DWT) [23], Bayesian optimization (BO) [24], and multilayer feedforward neural network (FFNN) to locate the dc-link faults.

A multilayer FFNN model based on BO is trained and evaluated using the selected features. BO is well-known as a powerful technique for optimizing black-box functions when closed-form expressions or surrogate models are unavailable [24]. A study in the literature found that BO provided a higher convergence rate than standard tuning methods after the neural network was adjusted [25,26]. The multi-stage BO introduced in this work reduces the computational burden by reducing the number of simulations needed to find the optimal design for any given neural network. As a result, it provides a well-tuned multilayer FFNN that can achieve improved response with better accuracy.

To simulate numerous fault scenarios, a four-terminal MT-HVdc system is developed in PSCAD/EMTDC, and the proposed model is investigated in a Matlab^®^ environment. Meanwhile, the proposed algorithm is compared to intelligent adversaries such as backpropagation neural networks (BP-NNs) and conventional FFNNs. The test results show that the suggested model performs with the highest fault localization accuracy.

Under the circumstances above, the main contribution of this study is:Our initial goal is to create a learning-based algorithm that relies on only one end of the communication link for fault location. Hence, eliminating reliance on the communication link.In general, a signal detected by a sensor is invariably interfered with by the surrounding environment or modified by the detecting equipment during the detection process, increasing failure chances. The DWT-based signal analysis model is used to eliminate interference from the observed signal to improve signal analysis and recognition.The energy or norm of the current and voltage signals at each frequency band gives a unique signature for different fault locations and has been found to be robust against noise. Therefore, it is used as an extracted feature for pattern recognition.The proposed algorithm must be able to locate internal faults with high fault impedances at further distances.

The remainder of the paper is organized in the following way.

Section 2 discusses the mathematical model derivation from a simple backpropagation algorithm to the improvised backpropagation algorithm. It also discusses the implementation of the proposed framework as well. Meanwhile, Section 3 introduces the conditions and properties of the chosen system model, which has been developed to capture fault data under dynamic fault scenarios. Section 4 covers the methodologies utilized to analyze input features extracted under dynamic fault scenarios. It also covers a denoising scheme that is used to denoise features before training and data preprocessing. Section 5 presents comparisons and analyses against adversaries. Finally, Section 6 concludes with a summary of the proposed algorithm. An overview of the proposed method is presented in the next section.

## 2. Proposed Framework

Figure 1 illustrates the architecture with the proposed methodology. During fault localization, the proposed method has two stages. In the first one, the captured fault window is filtered using discrete wavelets to rectify the noise issue, set to 10 ms. After denoising, the measured time-domain segment is transformed into a time-frequency domain by splitting it into low- and high-frequency components with a DWT-based multi-resolution analysis (MRA) technique.

The multilayer neural network receives decluttered information from voltage and current signals as inputs. It then estimates the fault site using the activation function. Levenberg–Marquardt backpropagation (LMBP) is implemented instead of a standard backpropagation algorithm with a performance function. It is a function of the ANN regression model and ground truth of fault sites. The Levenberg–Marquardt method is used to update the weight and bias. The Jacobian matrix of the performance function with respect to the weight and bias variables is calculated via the proposed backpropagation algorithm. After updating the weight and bias, the multilayer ANN is applied to determine the fault site. The multi-stage BO procedure is conducted prior to the update, aiming to increase accuracy during training and to provide an optimal multilayer FFNN by optimizing hyperparameters. The hyperparameters, unlike internal parameters (weights, bias, etc.), are set before the neural network is trained, and they influence the neural network’s performance. Regulating them via the trial-and-error method lengthens the training set-up time and may reduce accuracy. Hence, optimizing these hyperparameters enhances the accuracy and convergence speed [24]. In the following sub-section, the detailed architecture of the proposed algorithm is described.

### 2.1. Feedforward Neural Network (FFNN)

This study uses a feedforward neural network with a single hidden layer to model because a neural network with a single hidden layer can handle the most complex functions (i.e., one input layer, one output layer, and one hidden layer) [27]. In a multilayer FFNN, the basic building block is a neuron that mimics a biological neuron’s functions and behavior [27]. The schematic structure based on the neuron is shown in Figure 2.

Usually, a neuron has multiple inputs. Each element of the input vector *p* = [*p*_1_, *p*_2_, *K*, *p_R_*] is weighted by elements *w*_1_, *w*_2_, *K*, *w_j_* of the weight matrix W. Next, the bias of each neuron is summed with the weighted inputs to form the net-input *n*, expressed as:(1)$$n=\sum_{j=1}^{R}w_{j}p_{j}+b=W_{p}+b.\;$$

Following that, net-input *n* is sent via an activation function *f*, which results in the neuron’s output a. Mathematically expressed as:(2)a=f(n)

In this work, the activation function is based on the hyperbolic tangent sigmoid transfer function. The following equation presents it.
(3)$$f\left(x\right)=\frac{2}{1+e^{-2x\;}}-1\;$$

With reference to Figure 2, the multi-input FFNN executes the following equation:(4)$$a^{2}=f^{2}\left(\sum_{i=1}^{s}w_{1,i}^{2}f^{1}\left(\sum_{j=1}^{R}w_{i,j}^{1}p_{j}+b_{i}^{1}\right)+b^{2}\right)$$

The output of the neural network is represented by $a^{2}$. *R* stands for the number of inputs, the number of neurons in the hidden layer is denoted by *S*, and the *j*th input is represented by $p_{j}$. The activation functions of the output and hidden layers are represented by *f*^2^ and *f*^1^, respectively. The bias of the *i*th neuron is defined by $b_{i}^{1}$, whereas the bias of the neuron in the output layer is represented by $b^{2}$. The weight $\;w_{i,j}^{1}p_{j}$ represents the connection between the *j*th input and the *i*th neuron of the hidden layer. Meanwhile, the weight connecting the *i*th hidden layer source to the output layer neuron is denoted by $w_{1,i}^{2}$.

### 2.2. Backpropagation Algorithm

Following the definition of the FFNN, the next step is to create an algorithm for training such networks. To train the established multilayer FFNN, an error backpropagation algorithm based on the steepest descent technique is typically utilized [28]. For the proposed three-layer FFNN, we now express the function that represents the output of unit *i* in layer *m* + 1 as:(5)$$a^{m+1}=f^{m+1}\left(n^{m+1}\left(i\right)\right)$$

Then to propagate the function and generate net-input ($n^{m+1}\left(i\right)$) to unit *i*, the neuron in the first layer receives extracted features from the MT-HVdc system to provide an initial condition for Equation (5):(6)$$a^{0}=p$$

Equation (5) is further translated in matrix form for an *M* number of layers in a neural network as:(7)$$a^{m+1}=f^{m+1}\left(W^{m+1}a^{m}+b^{m+1}\right),\;\ldots ,\;m=0,\;1.$$
where $a^{m+1}$ and $a^{m}$ are the outputs of the network’s (*m*+1)th and *m*th layers. $b^{m+1}$ reflect the bias vector of the network’s (*m*+1)th layer. Here, external inputs passing to the network via Equation (7), the overall network’s outputs are equal to the outputs of the neurons in the last layer:(8)$$a=a^{\;M}$$

The objective of this study is to locate the dc-link faults. Therefore, the proposed multilayer FFNN requires a set of input–output pairs that characterize the behavior of an MT-HVdc system under faulty settings. Mathematically expressed as:(9)$$\left[\left(p_{1},t_{1}\right),\left(p_{2},t_{2}\right),\left(p_{3},t_{3}\right),\ldots ,\ldots ,\ldots ,\left(p_{Q},t_{Q}\right)\right],\;p_{q}\;\mathrm{is}\;\mathrm{input}\;\mathrm{and}\;t_{q}\;\mathrm{is}\;\mathrm{the}\;\mathrm{relevant}\;\;\mathrm{target}\;\mathrm{of}\;\mathrm{the}\;\mathrm{network}\;\mathrm{that}\;\mathrm{uses}\;\mathrm{for}\;\mathrm{training}.$$

After each input propagates through the multilayer FFNN during training, the network output is compared to the target. While doing so, the performance index for the backpropagation algorithm is the mean-square error (MSE), which is to be reduced by modifying the network parameters, given as:(10)$$F\left(x\right)=E\left[\left(e^{2}\right)\right]=E\left[{\left(t-a\right)}^{2}\right]$$

In the FFNN, x is the vector matrix containing the network weights and biases. However, in our case, the proposed network has multiple outputs. Therefore, Equation (10) generalized to:(11)$$F\left(x\right)=E\left[\left(e^{T}e\right)\right]=E\left[{\left(t-a\right)}^{T}\left(t-a\right)\right]$$

Since the steepest descent rule is utilized for the standard backpropagation algorithm, the performance index F(x) can be approximated as follows:(12)F^(x)=E[(t(k)−a(k))T(t(k)−a(k)))]=eT(k)e(k)

The squared error replaces the expectation of the squared error in Equation (11) at iteration step *k*. The steepest (gradient) descent algorithm for the estimated MSE is then:(13)wi,jm(k+1)=wi,jm(k)−∝dF^dwi,jm
(14)bim(k+1)=bim(k)−∝dF^dbim 

∝ is the learning rate, similar to the number of neurons (*S*); it is also a hyperparameter. Defined:(15)sim=dF^dnim
as the performance index (F^) sensitivity ($s_{i}^{m}$) that measures the changes in the net input of the *i*th element in layer *m*. Next, based on the chain rule, the derivate of Equations (13) and (14) using Equations (5), (12) and (15) can be simplified as:(16)dF^dwi,jm=dF^dnim∗dnimdwi,jm=sim∗ajm−1 
(17)dF^dbim=dF^dnim∗dnimdbim=sim 

Now with the definition of gradient, the steepest descent algorithm is approximated as:(18)$$w_{i,j}^{m}\left(k+1\right)=w_{i,j}^{m}\left(k\right)-\propto *s_{i}^{m}*a_{j}^{m-1}$$
(19)$$b_{i}^{m}\left(k+1\right)=b_{i}^{m}\left(k\right)-\propto *s_{i}^{m}$$

The following recurrence relation in matrix form can be satisfied by the sensitivity [29,30]:(20)$$s^{m}={\bar{F}}^{\;m}\left(n^{m}\right){\left(W^{m+1}\right)}^{T}s^{m+1},\;\mathrm{for}.\;m=M-1,\ldots ,\;2,1.$$

Equation (20) expresses the step used to propagate the sensitivities backward through a neural network. Mathematically, the sensitivities propagate backward across the network as:(21)$$s^{M}\to s^{M-1}\to \cdots \to s^{2}\to s^{1}$$
where
(22)$${\bar{F}}^{\;m}\left(n^{m}\right)=\left[\begin{array}{cccc} {\bar{f}}^{m}\left(n_{1^{m}}^{m}\right) & 0 & K & 0\\ 0 & {\bar{f}}^{m}\left(n_{2^{m}}^{m}\right) & & 0\\ M & M & & M\\ 0 & 0 & K & {\bar{f}}^{m}\left(n_{s^{m}}^{m}\right) \end{array}\right]$$

And
(23)$${\bar{f}}^{\;m}\left(n_{j^{m}}^{m}\;\right)=\frac{df^{m}\left(n_{j}^{m}\right)\;}{dn_{j}^{m}\;}$$

Whereas a recurrence relation is initialized at the final layer as:(24)$$s^{M}=-2{\bar{F}}^{\;m}\left(n^{M}\right)\left(t-a\right)$$

Now, we can summarize the overall backpropagation (BP) based on the steepest descent algorithm as (1): First, use Equations (6)–(8) to propagate the input through the network. (2): Next, using Equations (20) and (24), backpropagate the sensitivity. (3): Finally, using Equations (18) and (19), update the weights and biases.

### 2.3. Levenberg–Marquardt Backpropagation

The backpropagation algorithm exhibits asymptotic convergence properties while training the multilayer FFNN, which causes a slow convergence rate due to minor weight changes around the solution. Meanwhile, Levenberg–Marquardt (LM) backpropagation [29] is a variant of Newton’s method, which inherits the stability of the steepest descent algorithm and the speed of the Gauss–Newton algorithm [27,29,30]. Now, suppose we want to optimize performance index *F*(***x***); then, Newton’s method is:(25)$$x_{k+1}=x_{k}-A_{k}^{-1}\;g_{k}.$$
where ${A_{k}≅\nabla^{2}F\left(x\right)|}_{X=X_{k}}$, plus ${g_{k}≅\nabla F\left(x\right)\;|}_{X=X_{k}}$. Note that $\nabla^{2}F\left(x\right)$ represents the Hessian matrix, and ∇F(x) denotes the gradient. Let us assume that F(x) is a sum-of-squares function, then:(26)$$F\left(x\right)=\sum_{i=1}^{N}v_{i}^{2}\left(x\right)=v^{T}\left(x\right)v\left(x\right).$$

Then the gradient and Hessian matrix are expressed in matrix form as:(27)$$\nabla F\left(x\right)=2\;J^{T}\left(x\right)v\left(x\right).$$
(28)$$\nabla^{2}F\left(x\right)=2\;J^{T}\left(x\right)J\left(x\right)+2\;S\left(x\right).$$

J(x) denotes the Jacobian matrix as:(29)$$J\left(x\right)=\left[\begin{array}{cccc} \frac{dv_{1}(x)\;}{dx_{1}\;\;} & \frac{dv_{1}(x)}{dx_{2}\;\;} & \cdots & \frac{dv_{1}(x)}{dx_{n}\;\;}\\ \frac{dv_{2}(x)}{dx_{1}\;\;} & \frac{dv_{2}(x)}{dx_{2}\;\;} & \cdots & \frac{dv_{2}(x)}{dx_{n}\;\;}\\ M & M & \cdots & M\\ \frac{dv_{N}(x)}{dx_{1}\;\;} & \frac{dv_{N}(x)}{dx_{1}\;\;} & \cdots & \frac{dv_{N}(x)}{dx_{n}\;\;} \end{array}\right]$$
(30)$$S\left(x\right)=\sum_{i=1}^{N}v_{i}\left(x\right)\nabla^{2}v_{i}\left(x\right)$$

Assume that S(x)≈0, then Equation (30) (Hessian matrix) approximate as $\nabla^{2}F\left(x\right)≅2\;J^{T}\left(x\right)J\left(x\right)$**.** Next, Equation (25) updates after substituting Equation (27) and the approximation of Equation (28) as:(31)$$\Delta x_{k}=x_{k+1}-x_{k}=-{\left[\;J^{T}\left(x_{k}\right)J\left(x_{k}\right)\right]}^{-1}*\;J^{T}\left(x_{k}\right)v\left(x_{k}\right).$$

The matrix $(H=J^{T}J$
**)** may not be invertible using the Gauss–Newton method. This issue can be fixed by making the following changes to the approximation Hessian matrix:(32)G=H+μI

This modification to the Gauss–Newton method eventually leads to the LM algorithm [29]:(33)$$\Delta x_{k}=-{\left[\;J^{T}\left(x_{k}\right)J\left(x_{k}\right)+\mu_{k}I\;\right]}^{-1}\;J^{T}\left(x_{k}\right)v\left(x_{k}\right).$$

Now, using the $\Delta x_{k}$ direction, recalculate the approximated F(x). If a smaller number is obtained, then the computation procedure is repeated, but the parameter $\mu_{k}$ is divided by a factor (*α* > 1). If the value of *F*(***x***) does not decrease, then the value of $\mu_{k}$ for the next iteration in the step is multiplied by α.

The calculation of the Jacobian matrix is an essential step in the LM method. The elements of the Jacobian matrix are calculated using a slight modification to the BP algorithm to address the NN mapping difficulty [29]. For better understanding, similar to Equation (12) for the BP algorithm, Equation (26) is a performance index for the mapping problem in the LM algorithm, where the error vector is $v^{T}=\left[v_{1}\;v_{2\;}K\;v_{N}\right]$**=**$\left[e_{1,1}\;e_{2,1}\;K\;e_{s^{M},1}\;e_{2,1}K\;e_{s^{M},Q}\right]$, and the vector *x* parametric values are $x^{T}=\left[x_{1}\;x_{2\;}K\;x_{N}\right]$**=**$\left[w_{1,1}^{1},w_{1,2}^{1}\;K\;,w_{S{,}^{1}R}^{1}\;.\;b_{1}^{1},\;K\;,\;b_{S^{1}}^{1}.w_{1,1}^{2}\;,\;K\;,b_{S^{M}}^{M}\right]$, subscript *N* defined as *N* = *Q* ∗ $S^{M}$.

Similarly, the *n* subscript is defined as *n* = $S^{1}\left(R+1\right)+S^{2}\left(S^{1}+1\right)+\ldots +S^{M}\left(S^{M-1}+1\right)$ in the Jacobian matrix. Now making all these substitutions in Equation (29) of the Jacobian matrix as:(34)$$J\left(x\right)=\left[\begin{array}{cccccc} \frac{de_{1,1}\;\;}{dw{\;}_{1,1}^{1}\;\;} & \frac{de_{1,1}\;\;}{dw{\;}_{1,2}^{1}\;\;} & \cdots & \frac{de_{1,1}\;\;}{dw{\;}_{S^{1},R}^{1}\;\;} & \frac{de_{1,1}\;\;}{db{\;}_{1}^{1}\;\;} & \cdots \\ \frac{de_{2,1}\;\;}{dw{\;}_{1,1}^{1}\;\;} & \frac{de_{2,1}\;\;}{dw{\;}_{1,2}^{1}\;\;} & \cdots & \frac{de_{2,1}\;\;}{dw{\;}_{S^{1},R}^{1}\;\;} & \frac{de_{2,1}\;\;}{db{\;}_{1}^{1}\;\;} & \cdots \\ M & M & & M & M & \\ \frac{de_{S^{M},R}\;\;}{dw{\;}_{1,1}^{1}\;\;} & \frac{de_{S^{M},R}\;\;}{dw{\;}_{1,2}^{1}\;\;} & \cdots & \frac{de_{S^{M},R}\;\;}{dw{\;}_{S^{1},R}^{1}\;\;} & \frac{de_{S^{M},R}\;\;}{db{\;}_{1}^{1}\;\;} & \cdots \\ \frac{de{\;}_{1,2}\;\;}{dw{\;}_{1,1}^{1}\;\;} & \frac{de{\;}_{1,2}\;\;}{dw{\;}_{1,2}^{1}\;\;} & \cdots & \frac{de{\;}_{1,2}\;\;}{dw{\;}_{S^{1},R}^{1}\;\;} & \frac{de{\;}_{1,2}\;\;}{db{\;}_{1}^{1}\;\;} & \cdots \\ M & M & \cdots & M & M & \cdots \end{array}\right]$$

Until now, the standard BP algorithm has been used to calculate the Jacobian matrix terms as follows:(35)dF^(x)dxI=deqTeqdxI

Meanwhile, in the LM algorithm, the terms for the elements of the Jacobian matrix can be calculated using the following:(36)$${\left[J\right]}_{h,I}=\frac{dv_{h}}{dx_{I}}=\frac{de_{k,q}}{dw_{i,j}}$$

Thus, rather than computing the derivatives of the squared errors as in standard backpropagation, we are calculating the derivatives of the errors in this modified Levenberg–Marquardt algorithm. Similar to the concept for standard backpropagation sensitivities, a new Marquardt sensitivity is defined as follows:(37)[J]h,I=dek,qdwi,jm=dek,qdni,jm∗dni,jmdwi,jm =si,h^m∗aj,qm−1
if $x_{I}$ is a bias,
(38)[J]h,I=dek,qdbim=dek,qdni,qm∗dni,qmdbim =si,h^m

As previously stated, the Marquardt sensitivity can be determined using the same recurrence relation as the standard sensitivities. However, toward the conclusion of the final layer, there is only one modification for calculating the new Marquardt sensitivity:(39)si,h^M=dek,qdni,qM=d(tk,q−ak,qM )dni,qM=dak,qM dni,qM
for i=k , it is
(40)si,h^M=−f^M(ni,qM)
for i≠k, it is equal to zero. Note that f^M and its matrix can be defined with the help of Equations (22) and (23). In the proposed model, when extracted features from the MT-HVdc network are applied to the multilayer FFNN as an input ($p_{q})$ and the corresponding output ($a_{q}^{M})$ is processed, the LMBP algorithm is initialized with the following:(41)Sq^M=−F^M(nqM)

Each column of the matrix in Equation (41) is a sensitivity vector that must propagate back through the network to generate one row of the Jacobian matrix. The columns are propagated backward as follows:(42)Sq^m=F^m(nqm)(Wm+1)Sq^m+1

The augmentation that follows then obtains all of the Marquardt sensitivity matrices for the overall layers.
(43)S^m=[S1^m⋮S2^m⋮S3^m⋮S4^m⋮K⋮SQ^m]

The proposed algorithm based on Levenberg–Marquardt’s backpropagation algorithm for fault allocation is given for clarity in Table 1.

### 2.4. Parameter Optimization

Hyperparameters should be distinguished from internal parameters such as weights and biases that are taken into account by the Levenberg–Marquardt backpropagation algorithm in the FFNN model. However, finding values for hyperparameters is a non-convex optimization process for optimal fitting. This is because, like the MT-HVdc system, most existing systems do not have linear responses to their control parameters. From the standpoint of optimization, the problem can be presented as follows:(44)$$\min_{x\epsilon X^{d}}f\left(x\right),\;\;\;where\;x\epsilon X\subset ℜ$$

x is the input vector (control parameters) of dimension d. f(x) is an objective function that depicts a multiscale system with high dimensional control parameters functioning under high-speed channels, such as an FFNN-based relaying model under dynamic conditions to protect the MT-HVdc grid. It is not a simple task to create a precise and accurate model of such systems in this situation. As a result, it is necessary to approach the problem in Equation (44) using the black-box settings shown in Figure 3.

#### 2.4.1. Black-Box Settings

In most black-box systems, including MT-HVdc grids relaying models, it is not easy to acquire f(x) gradient information at an arbitrary value of x. However, gradient information is not required when employing BO based on Gaussian processes (GPs) [31]. As a result, it is a promising and appropriate candidate for black-box optimization. While optimizing, BO is an active learning method that chooses the next observation to maximize the reward for solving Equation (45). Its foundation is Bayes’ Theorem.
(45)$$P\left(f|D_{1:t}\right)\propto P\left(D_{1:t}|f\right)P\left(f\right)$$

P(f), $P\left(f|D_{1:t}\right)$ and $P\left(D_{1:t}|f\right)$ are probabilities of prior, posterior, and likelihood based on the current observations, i.e.,$\;D_{1:t}=\left[\left(x_{1},y_{1}\right),\left(x_{2},y_{2}\right),..,\left(x_{t},y_{t}\right)\right]$. Various predictive and distributional models can be used as priors in BO, but the GP is preferred due to its practical and theoretical advantages [31].

#### 2.4.2. Gaussian Process (GP)

In the GP, the surrogate model replicates the behaviors of the expensive underlying function. While doing this, the underlying function f(x) that requires optimization is represented in BO as a collaborative and multidimensional Gaussian process. The mean (μ) and covariance (K) functions are calculated using:(46)$$f_{1:t}=N\left(\mu \left(x_{1:t}\right),K\left(x_{1:t}\right)\right)$$

In BO, Equation (46) illustrates the process in which the predictive GP is trained. It is worth noting that, unlike other machine-learning algorithms, the goal of BO is to properly forecast where global extrema are situated in the sample space based on previous observations rather than to develop predictors that cover the entire sample space. Furthermore, the problem in Equation (44) is solved using black-box settings, implying that we do not have any prior information about the underlying function. Therefore, to improve the regression quality of the GP, we use a popular kernel/covariance function called the automatic relevance determination Matern 5/2 function in conjunction with a zero-mean GP for P(f), given as:(47)$$K\left(x\right)=\left[\begin{array}{ccc} k\left(x_{1},x_{1}\right) & \cdots & k\left(x_{1},x_{t}\right)\\ \vdots & \ddots & \vdots \\ k\left(x_{t},x_{1}\right) & \cdots & k\left(x_{t},x_{t}\right) \end{array}\right]$$
(48)$$k\left(x_{i},x_{j}\right)=\sigma_{f}^{2}\left(1+\sqrt{5r}+\frac{5}{3}r^{2}\right)e^{-\sqrt{5r}}\;$$
where r= $\left(\sum_{d=1}^{D}{\left(x_{i,d}-x_{j,d}\right)}^{2}/\sigma_{d}^{2}\right){)}^{1/2}$; $\sigma_{f}$ and $\sigma_{d}$ are hyperparameters of K(x). These hyperparameters are modified throughout the training phase to reduce the GP’s negative-log marginal likelihood using the global or local method. Each parameter in an ARD-type kernel has a scaling parameter that must be set. If the $\sigma_{d}$ of one parameter is larger than the others after the GP-based predictive model has been trained, then it can be assumed that a change in this parameter has less sensitivity on the prediction. Furthermore, if a certain parameter has a greater effect, then the proposed solution in BO will alter the training process to reduce $\sigma_{d}$ of that parameter in comparison to others. These advantages make the underlying function more interpretable and serve as an implicit sensitivity analysis.

#### 2.4.3. Acquisition Function

Since the original function f(x) is hard to estimate, based on a predefined strategy and auxiliary optimization, an acquisition function u(x) is obtained to find the next point $x_{t+1}$ of the solution. It is worth noting that u(x) does not require any additional points; instead, it relies on past sample knowledge to make predictions at candidate points.
(49)$$\mu \left(x_{t+1}\right)=k^{T}K^{-1}f_{1:t}$$
(50)$$\sigma^{2}\left(x_{t+1}\right)=k\left(x_{t+1},x_{t+1}\right)-k^{T}K^{-1}k$$

Then predictive distribution at the next point is given as:(51)$$P\left(f_{t+1}|D_{1:t},x_{t+1}\right)\sim N\left(\mu \left(x_{t+1}\right),\sigma^{2}\left(x_{t+1}\right)\right)$$

The most prominent acquisition functions in BO are the probability of improvement, upper confidence bound and expected improvement per second. However, we propose an expected improvement per second-plus in this paper. In comparison, it allows for faster model building and optimization, and the term ‘plus’ prevents a region from overexploiting (more search for a global minimum). Expected improvement (EI) is given as:(52)μ(EI)=(μ(x)−f^∗−ζ )Φ(Z)+σ(x)ϕ(Z)
where f^∗ is the best point observed so far. ζ is a hyperparameter for μ(EI), Z=(μ(x)−f^∗−ζ)/σ(x), ϕ(.) and Φ(.) are the probability density function and cumulative distribution function of normal distribution. Further interpreted in EI per second (*EIpS*) as:(53)$$EIpS\left(x\right)=\mu \left(EI\right)/\mu_{S}\left(x\right)$$
where $\mu_{S}\left(x\right)$ is the posterior mean of the timing Gaussian process model, respectively. The next sampling point $x_{t+1}$ is found by minimizing the expected improvement per second-plus EIpSp(x) acquisition function.
(54)$$x_{t+1}=argmin\;EIpSp\left(\mu \left(x_{t+1}\right),\sigma^{2}\left(x_{t+1}\right)\right)$$

In doing so, the proposed acquisition function escapes the local objective function minimum and searches for a global minimum by setting $\sigma_{f}\left(x\right)$ to be the posterior objective function ($P\left(f|D_{1:t}\right)$) standard deviation at point x. Let $\sigma_{NP}$ be the additive noise posterior standard deviation so that $\sigma_{Q}^{2}\left(x\right)=\sigma_{F}^{2}\left(x\right)+\sigma_{NP}^{2}$. The positive exploration ratio is denoted by $t_{\sigma NP}$. After each iteration, the acquisition function evaluates if the next point x satisfies $\sigma_{f}\left(x\right)<t_{\sigma NP}\sigma_{NP}$. If this is the case, then the acquisition function will announce that x is overexploiting and adjust its kernel function by multiplying θ by the number of iterations [32]. When compared to EIpS(x), this adjustment increases the variation $\sigma_{Q}$ for points between observations. It then creates a new point using the newly fitted kernel function. However, if the new point x is still being overexploited, then the function multiplies θ by a factor of ten and tries again. This process is repeated five times, with the goal of generating a point x that is not overexploited. The new x is accepted as the next exploration ratio by the proposed acquisition function. As a result, it manages the tradeoff between examining new points, searching for a better global solution, and focusing on nearby already investigated points. The whole process optimizes the FFNN structure in a much faster and more efficient manner with a reduced computation burden.

#### 2.4.4. Implementation of Proposed Framework

The steps to train the FFNN model with the LMBP algorithm and optimize network hyperparameters with the Bayesian algorithm are demonstrated in Figure 4.

In step 1, fault location and impedance are modified to create the training and testing datasets for several simulations. Additionally, data events are labeled and normalized according to criteria to improve the training process in this mode. The ANN hyperparameters are determined by feeding the training dataset into BO’s AI model until the maximum number of iterations is reached. The AI model is updated each time the maximum number of iterations is reached. In step 2, the optimal hyperparameters of the ANN, which gives the minimum root-mean-square error (RMSE), are selected by BO, and the FFNN is trained for the given training data with the help of the LMBP algorithm. In step 3, the trained ANN model is evaluated on a different testing dataset from the training dataset.

To prevent overfitting, K-fold cross-validation was used during the assessment with K = 5. RMSE =$\sqrt{\frac{1}{R_{z}}\sum_{n=1}^{R_{z}}\left(y_{n}-y_{n}^{\circ }\right)},$ *R_z_* stands for the data size, *y_n_* for the actual output, and $y_{n}^{\circ }$ represents the predicted output. The proposed framework can now be implemented; a system model will be presented in the next section, which enables the collection and analysis of input features for fault types, matching the theoretical foundation to real-world fault scenarios, and using intelligent computation to train and evaluate the framework’s effectiveness.

## 3. System Model

The electrical power from two offshore wind farms is transferred to two onshore converters through dc transmission, as shown in Figure 5 [33]. A boundary is defined by installing current limiter inductors at the end of a dc line. Other test grid settings and MMC parameters are provided in Table 2 and Table 3. The cable specifications are provided in Table 4. It is a single-end scheme, which means that information will be gathered near circuit breakers and inductor lines.

### Model Output

As shown in Table 5, the examined system model has several outputs that can be used to determine fault distance from a relay contact point. Additionally, it shows fault resistances and fault types along with a total of 714 dc-link fault scenarios (*k*) for training. By doing so, dc-link faults are categorized into pole-to-pole (PTP) and pole-to-ground (PTG) faults. It is important to note that a dc-link problem is an internal fault, so the criteria $\left(dV_{dc}/dT\right)$ should be applied when an internal failure occurs. By activating this criterion, the trained algorithm begins sampling relevant values for the 10 ms time window and estimating the fault distance. Note that the fault detection strategy is selective in nature.

## 4. Data Processing

The fact that the initial travelling waves of the voltage and current induced by the dc-link faults from the system above contain helpful information about fault distance is exploited in this study [7]. However, noise interference is expected, considering the dynamic disturbances associated with the MT-HVdc system. Therefore, the following sub-section discusses the noise suppression mechanism before processing data for the regression model.

### 4.1. Signal Processing

The implementation of the DWT to suppress noises from a measured signal is shown in Figure 6 [34].

#### 4.1.1. Setting Numbers of Decomposition Layers

Transforming discrete wavelets into more decomposition layers helps separate noise from the original signal, resulting in better signal filtering. We have chosen eight levels to keep the balance between signal processing burden and robustness against noise, corresponding to the frequency band of 195.3–390.6 Hz at a sampling frequency of 50 kHz.

#### 4.1.2. Selection of Mother Wavelet Function

The next critical step in the denoising scheme is choosing a mother wavelet. A literature review and practical results presented in the previous studies show that Daubechies (dB) is an appropriate mother wavelet for analyzing fault signals [35]. It is suggested that, in this study, the Pearson correlation coefficient be used to determine the correlation between the Daubechies wavelet function and the cable fault signals in order to determine the best mother wavelet function. The mother wavelet function is written as follows:(55)$$\emptyset =\sum^{\;}\left(X-\bar{X}\right)\left(Y-\bar{Y}\right)/\sqrt{\sum^{\;}{\left(X-\bar{X}\right)}^{2}{\left(Y-\bar{Y}\right)}^{2}}\;\;$$
where X is the original fault signal, $\bar{X}$ denotes the original fault signal’s average, Y denotes the noise-eliminated fault signal, and $\bar{Y}$ denotes the noise-eliminated fault signal’s average.

#### 4.1.3. Set the Threshold and Filter the Signal

After selecting the mother wavelet, the noise from the fault signal can be filtered out. The Universal threshold is multiplied by the median of each decomposition layer after wavelet decomposition to automatically set the threshold, as expressed:(56)$$\lambda_{j}=\frac{\sigma_{j}}{0.6745}*\sqrt{2\log n_{j}}$$

$\lambda_{j}$ is the threshold of the jth decomposition layer, $\sigma_{j}$ is the median of the jth decomposition layer, and $n_{j}$ is the signal length of the jth decomposition layer. After setting the threshold, the noise is filtered out through the thresholding process. This thresholding process usually includes soft and hard thresholds [35]. However, in this study, a hard threshold is set to filter out the noise.
(57)$$\delta_{\lambda }{}^{Hard}=[\begin{array}{c} x\left(t\right),\;\;\;\;\;if\left|x\left(t\right)\right|>\lambda \;\;\;\;\;\;\;\;\;\;\;\;\;\;\;\;\\ 0,\;\;\;\;\;\;\;\;\;\;\;\;\;\;\;\;\;\;\;\;otherwise \end{array}$$

This equation demonstrates that the hard threshold retains a larger wavelet coefficient while the coefficient below the threshold is set to zero. Finally, using inverse DWT (IDWT), the signal processed by the hard threshold can be configured layer by layer into a noise-free signal. The implementation of the proposed denoising approach with a 20 dB signal-to-noise ratio (SNR) is shown in Figure 7.

### 4.2. Feature Extraction Set-Up

After selecting and denoising the signal, the feature extraction stage is critical for data-driven-based fault detection and location estimation problems. Extracted features are measurable data taken from the transient of the current- and voltage-filtered signals to create a feature vector. This feature vector should be dimensionally compact to successfully implement the learning and generalization processes in the estimation algorithms for fault location. The feature extraction stage is divided into two sub-stages. The first stage involves decomposing all generated samples for each fault location up to eight levels using DWT-MRA to obtain wavelet coefficients. The wavelet coefficients are *Aj* approximation and *Dj* detail levels. For each type of fault location, vectors of D1–D8 and A8 coefficients are obtained. The second stage of feature extraction involves providing effective and appropriate statistical parameters for feature vector creation to reduce the collected data and improve estimation performance.

#### 4.2.1. Feature Extraction Results

When a large number of high-frequency components of voltage and current signals are fed for training, several learning tools face problems due to a limitation on the input space dimension. These learning tools lack the capability to provide suitable learning patterns with a large number of features. This is due to the enlargement of the structure and an extreme increase in the number of learning parameters [11]. The regression model used in this study is designed to train with the second norm (referred to as the norm) of the wavelet coefficients. In general, the decomposed signal’s norm for wavelet coefficients is determined as follows:(58)$$norm_{D_{j}}=\sqrt{\sum_{i=1}^{n}{\left|\;D_{i,j}\right|}^{2}}$$
(59)$$norm_{A_{j}}=\sqrt{\sum_{i=1}^{n}{\left|\;A_{i,j}\right|}^{2}}$$

j denotes the decomposition level, and the maximum level of decomposition is *N*. The detail and approximate coefficients have n values at level j. Overall, the proposed energy vector obtained from the MRA-based DWT for any current or voltage signal from a given time window is represented as
(60)$$x=\left[\;norm_{D_{1}},norm_{D_{2}}\;,\ldots ,\;norm_{D_{8}},norm_{A_{8}}\right]$$

Using the MRA-based DWT, norm values of current for ground faults at various sites are calculated and presented in Figure 8, respectively. There is a distinct difference in the approximate norms between the given fault locations at levels D6 through D8. These differences in norms indicate that the obtained features contain distinct fingerprints for estimating ground faults at various places. Figure 9 shows the obtained features of the voltage signal for ground faults between locations 40 to 200 km.

In Figure 9, the norm values for each location are significantly different in the dominant frequency band between D5 and D8 and can be used as input vectors to establish fault estimation rules. Similarly, as illustrated in Figure 10, a unique signature of the pole-to-pole fault may be derived at different frequency bands. A schematic diagram for the feature vector development process is shown in Figure 11.

#### 4.2.2. Training Set-Up

Following preprocessing strategies, these extracted features are standardized for computational simplification. The decluttered training dataset is then applied to the BO-based AI model to find the appropriate hyperparameters for the FFNN once the feature vectors have been determined. The input vector **p** = (x_1_, x_2_, x_3_, x_4_) of 10 ms is designed for the FFNN input; two inputs (x_1_, x_2_) represent the transient dc current second norm from positive and negative poles, while the rest (x_3_, x_4_) indicate the dc voltage second norm from positive and negative poles. This corresponds to 36 inputs for each training sample (total training samples = ***k*** = 714). In doing so, BO’s AI model is modified each time until the maximum number of iterations is reached. BO then selects the ideal FFNN hyperparameters that result in the lowest RMSE, and the FFNN is trained using the LMBP algorithm. The final RMSE obtained is 0.0132, with a total evaluation time of 39.3428 s for 30 iterations. Some key hyperparameters of the multilayer FFNN model obtained via BO are presented in Table 6.

## 5. Simulation Results and Discussions

A.Metric for Evaluation and Testing Set-Up

Although, during validation, the selected models’ average estimation accuracy was 98.94%. However, we tested our method for further investigation using case studies given in Table 7. For verification and more in-depth analysis, a performance index based on percentage error was used as follows:(61)$$\mathrm{Percentage}\;\mathrm{error}=\frac{\mathrm{Actual}\;\mathrm{Location}-\mathrm{Prediction}\;\mathrm{location}}{\mathrm{Total}\;\mathrm{lenght}\;\mathrm{of transmission}\;\mathrm{line}}\times 100$$

### 5.1. Case 1 (Fault Location)

In Case 1 (under varying fault locations and fault resistance), the functionality of the proposed technique was tested using the scenarios given in Table 7. After thorough training, fault analyses were carried out with varying fault distances and resistances. Table 8 shows the 800 test samples, absolute and percentage errors for two types of dc-link faults: PTP and PTG. It can be observed that the percentage error for the testing dataset was found to be 0.4927% and 0.5361% for the PTP fault and PTG fault, respectively. The proposed technique’s total percentage error was found to be 0.5144 percent, which demonstrated that the misclassification was well within acceptable bounds.

In addition, Figure 12 depicts the percentage inaccuracy for the proposed technique in locating PTP faults on line 13 PTP faults with fault distances ranging from 5 km to 200 km. With a maximum percentage error of 1.3174% at 175 km and a minimum value of 0.00103% at 15 km, the findings revealed that the proposed algorithm had no major impact on the variance of fault distance. Therefore, the proposed approach is suitable for locating close-in and far-away faults.

### 5.2. Case 2 (Fint)

Apart from fault location, it is important to note that the characteristics and amplitude of faulty signals, such as voltage and current measured at the local terminal, are also determined by fault parameters such as fault resistance. Therefore, it is crucial to highlight the proposed approach’s performance under diverse fault resistances. This section analyzes the proposed algorithm’s performance for in-depth fault resistance validity ranging from 10 to 385 Ω, and the results are given in Table 9. Notably, in the event of high fault resistance, such as 385 Ω, with an actual fault distance of 185 km, the energy of the travelling waves tended to be on the lower side, bringing the system closer to the steady state. However, the proposed algorithm with selected features extracted even the most minute voltage and current information. For example, the predicted fault distances for PTP and PTG at 385 Ω were 183.63147 km and 186.93141 km, respectively. The associated misclassification of 0.68427% and 0.96571% for each fault type was well within acceptable limits.

### 5.3. Case 3 (Noisy Events)

In this case, a white Gaussian was added to the testing signals to examine the proposed fault-locating scheme under various noisy occurrences. Original signals with SNRs ranging from 20 to 45 dB were employed to assess fault location performance. Table 10 indicates that the proposed scheme could locate all sorts of faults with a reasonable mean percentage error rate for close-in, mid-point of line, and far-end of line. In the case of 45 dB noise additions at the far end of 155 km of the dc-link, the total mean percentage error was 0.72424% and 0.83147% for PTP and PTG faults. It is worth noting that the proposed method was noise-resistant because of the denoising process with better threshold settings and functions. This improved the estimation accuracy despite the high noise level of 20 dB with an overall mean percentage error of 0.9411% and 0.8561% for PTP and PTG faults, respectively.

### 5.4. Case 4 (Comparison with Existing Methods)

To further validate the proposed scheme’s robustness, Figure 13 replaces it with intelligent adversaries such as the conventional FFNN and BP-NN with an original current signal as the input under the testing conditions listed in Table 7.

On a dual 2.9 GHz, Intel Core i7 with 16 GB RAM, the current version of the algorithm implemented in Matlab^®^ R2020a took 39.3428 s to run. Thirty ANN models were selected, trained, and validated with this runtime. It was approximately five times faster than a conventional FFNN configured manually with hyperparameters. The results showed that the proposed algorithm performed better than the BP-NN and had the lowest percentage error (i.e., 0.49%, 0.54% and 0.51%) for all fault types. In terms of percentage error, the conventional FFNN with hyperparameters such as 15 neurons in the hidden layer and a learning rate of 0.01 gave an average percentage error of 0.56%. This showed that efficient features and regulating parameters in the proposed algorithm helped to increase the interpretability of the spectrum generated by the wavelet.

## 6. Comparison and Analysis

This section compares the proposed methodology with existing fault estimation schemes for the MT-HVdc grid.

### 6.1. Non-AI-Based Methods

The proposed fault location method utilizes a continuous wavelet transform on dc line current signals in the MT-HVdc network [36]. The technique is quite efficient; however, a high sampling frequency of 200 kHz and time-synchronized measurements are required. Further, evaluation under high fault resistance has not been investigated thoroughly. Another work used time-stamped measurements to locate faults at a 200 kHz sampling frequency [37]. The proposed model is robust against noise measurement, but high sampling frequency and synchronized measurements could be a barrier to practical applications. The single-ended TW-based fault location model has no synchronized measurement issue [38] but has a high sampling frequency (100 kHz) [39]. In another example, modal voltage and current measurements are sampled at 1 MHz to develop a single-end fault location model [40]. However, it has only been tested for 100 Ω fault resistance. All the aforementioned TW-based fault location models require a high sampling frequency for good accuracy. Such a requirement is frequently considered a drawback. In comparison, the proposed single-end fault location approach operates with reasonable sampling frequency and tests against fault resistance as high as 485 Ω.

### 6.2. AI-Based Methods

Among the fault location approaches, learning-based techniques fall into a distinct category. Even though such practices are commonly utilized in AC systems for fault localization, few papers discuss their relevance to MT-HVdc networks. For example, an extreme learning machine was proposed to locate the fault in the MT-HVdc network [41]. Voltage and current measurements were captured at a 500 kHz sampling frequency during the learning phase to perform the wavelet transform and s-transform for feature extraction. However, the entire scheme has been tested for fault resistance up to 100 Ω. Similarly, the high voltage and current measurements sampled at 200 kHz and the investigation of highly resistive faults are missing [42]. Another method applied a traditional two-ended TW-based fault location algorithm to current measurements sampled at 5 kHz [43]. The distance inaccuracy caused by the moderate sampling frequency was subsequently reduced using a machine-learning approach. However, utilizing multiple distributed sensors on long transmission added cost to the method. With the help of the ANN, the real-time implementation of the proposed method is quite efficient. It has been proven to have a low execution time on low-spec machines [44]. Further, all the aforementioned models do not discuss the optimization of the machine-learning model. The proposed approach optimizes the pre-training set-up with the help of Bayesian optimization.

## 7. Conclusions

At first, a novel dc fault location scheme based on AI for a meshed dc grid is proposed. The BO-based FFNN model with DWT application is used to determine the best hyperparameters that improve the selected model’s performance while keeping the RMSE low. Levenberg–Marquardt backpropagation is used to adjust weights and biases during training for the chosen multilayer FFNN model. The contribution of this work is summarized as follows:The wavelet coefficient energies of voltage and current over 10 ms are calculated and denoised during the learning phase for feature extraction. This leads to fewer features yet is robust for the learning model.A comprehensive training dataset is collected to train the multilayer FFNN model for different fault locations by varying fault impedance.The performance of this model is then evaluated on data points that are not included in the training dataset. The study results show that the fault location can be calculated using the FFNN for fault resistance up to 485 Ω.Because the signal and Gaussian noise are integrated into the FFNN training sets, the influence of the noise-contained environment is reduced.Due to plug-and-play capability, the suggested intelligent algorithm is tailored for a multi-vendor-based fault location estimation strategy in meshed MT-HVdc grids.The case studies show that the proposed scheme performs well against many variables, such as different fault resistances, transmission line lengths, and non-ideal noise events. Thus, that makes it feasible for practical application in the MT-HVdc grid.

In future work, variable time windows will be used to consider the effect of the fault location, fault resistance, and computational burden. This work provides an analysis of the fault location estimation method for HVdc cable grids that can be applied to hybrid cable–overhead line systems as well.

## Figures and Tables

**Figure 1 sensors-22-09936-f001:**
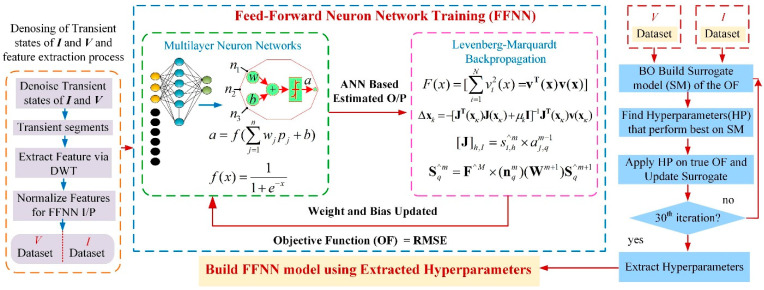
Proposed architecture.

**Figure 2 sensors-22-09936-f002:**
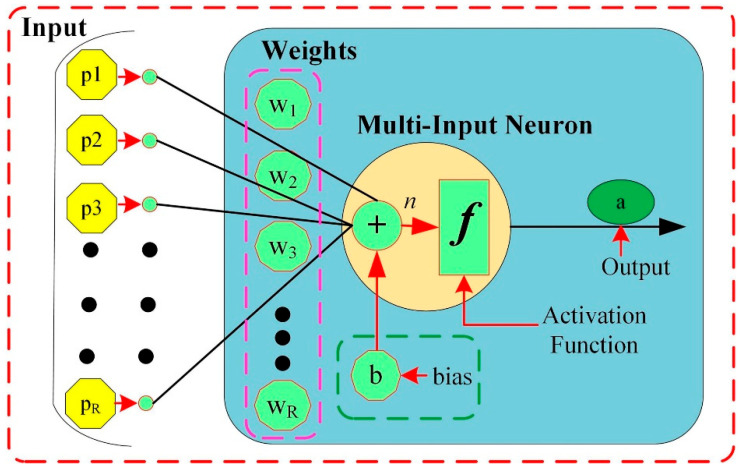
Structure of the multi-input neuron.

**Figure 3 sensors-22-09936-f003:**
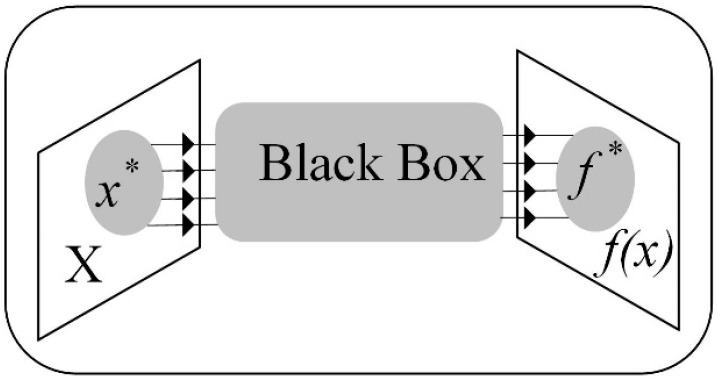
Optimization of black-box systems.

**Figure 4 sensors-22-09936-f004:**
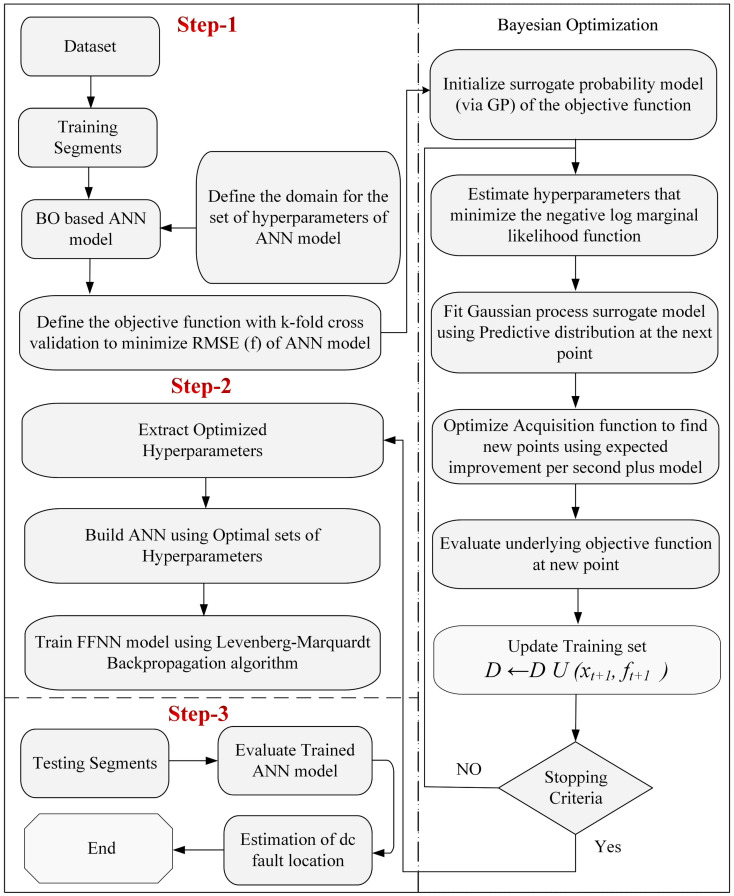
Proposed Framework.

**Figure 5 sensors-22-09936-f005:**
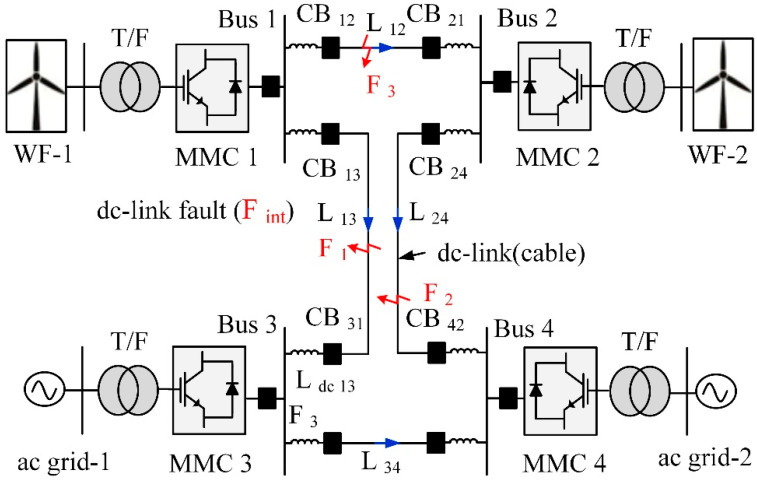
Configuration of MMC-based dc grid.

**Figure 6 sensors-22-09936-f006:**
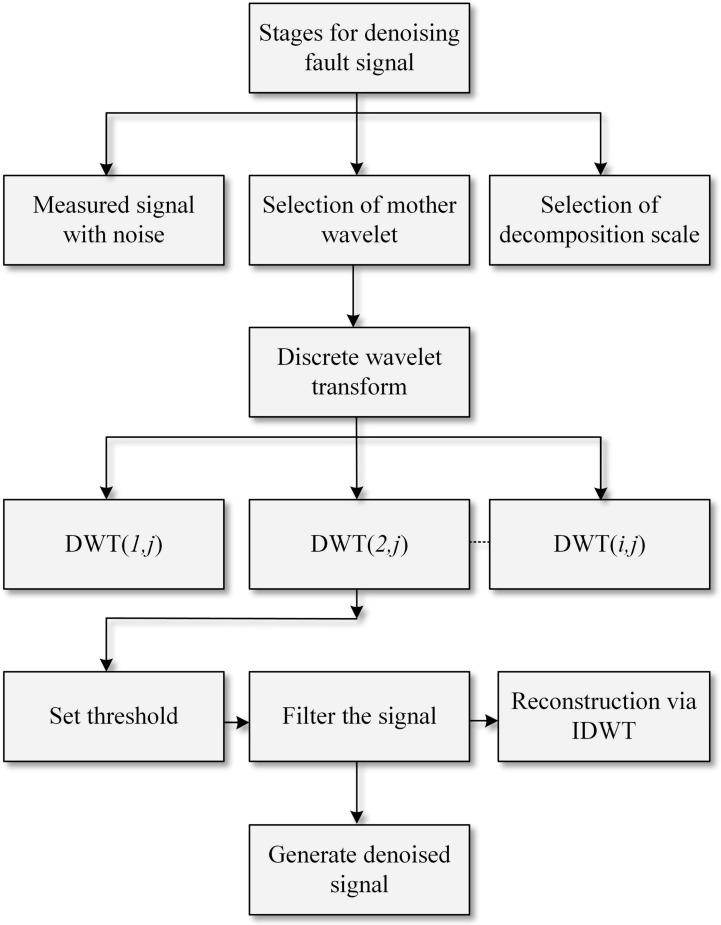
Signal denoising.

**Figure 7 sensors-22-09936-f007:**
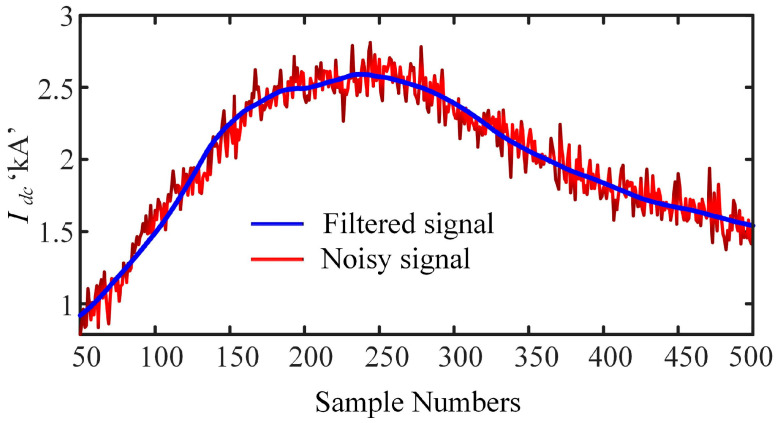
Effect of the denoised solution on the contaminated signal of 20 dB signal-to-noise ratio (SNR).

**Figure 8 sensors-22-09936-f008:**
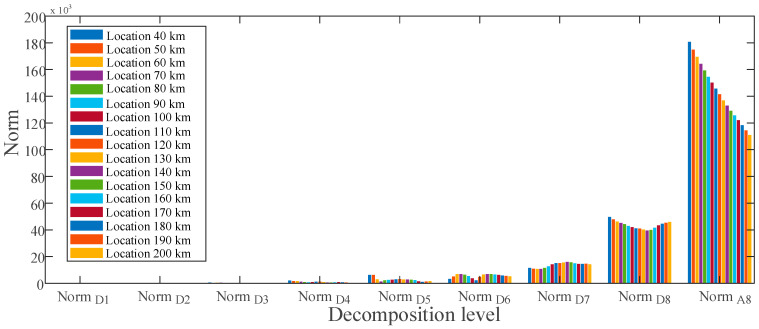
Feature vector extracted for ground fault at various locations of the current signal.

**Figure 9 sensors-22-09936-f009:**
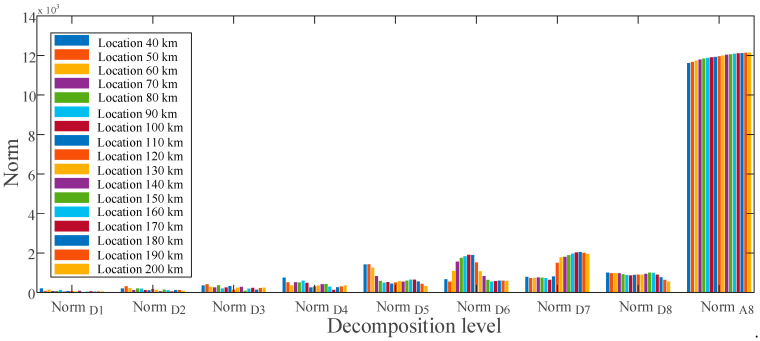
Feature vector extracted for ground fault at various locations of the voltage signal.

**Figure 10 sensors-22-09936-f010:**
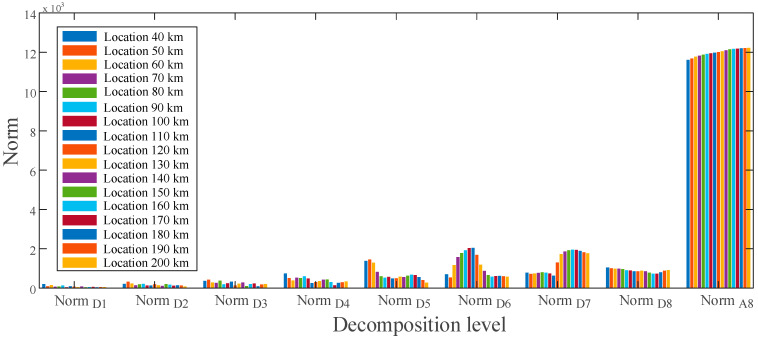
Feature vector extracted for the PTP fault at various locations of the voltage signal.

**Figure 11 sensors-22-09936-f011:**
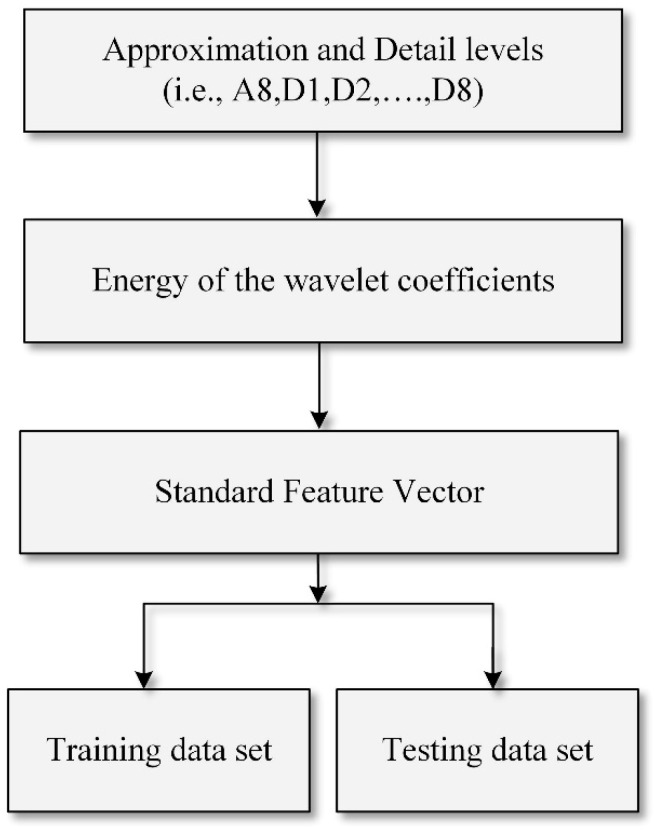
Flow chart for the development of the feature vector.

**Figure 12 sensors-22-09936-f012:**
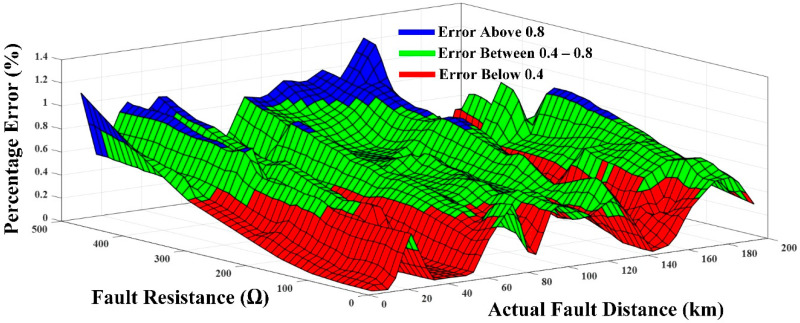
Accuracy of the proposed technique.

**Figure 13 sensors-22-09936-f013:**
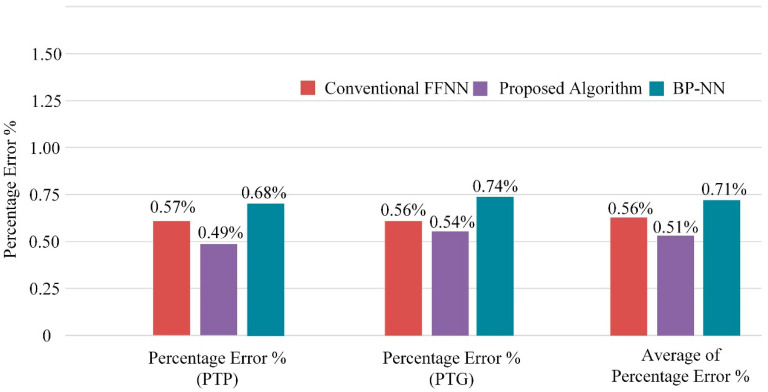
Comparative analysis.

**Table 1 sensors-22-09936-t001:** LMBP algorithm.

LMBP Algorithm for the Fault Location Process
With initial weights and bias (randomly generated), all extracted features should be fed into the FFNN as inputs. The outputs of the corresponding features are computed in the network using Equations (6) and (7), followed by error prediction using $e_{q}$ = $t_{q}-a_{q}^{M}$. Using $F\left(x\right)=\sum_{q=1}^{Q}{\left(t_{q}-a_{q}\right)}^{T}(t_{q}-a_{q})$, calculate the sum of squared errors for all inputs with the *Q* targets in the training set. After initializing with Equation (41), calculate the sensitivity using Equation (42) and augment the individual matrices into the Marquardt sensitivities using Equation (43). Meanwhile, Equations (37) and (38) are used to determine elements of the Jacobian matrix. Then, to obtain $\Delta x{\;}_{k}$, update Equation (33) to adjust weights and biases. Using $x{\;}_{k}+\Delta x{\;}_{k}$ recalculate the total of the squared errors. If the newly generated error value is less than the previous one, then divide $\mu_{k}$ by *α* and return to step a with $x{\;}_{k+1}=x{\;}_{k}+\Delta x{\;}_{k}$. If the recalculated value does not decrease, then multiply $\mu_{k}$ by *α* and return to step c with the new weights.

**Table 2 sensors-22-09936-t002:** Converter parameters.

Station	Rated dc Voltage [kV]	Rated Capacity [MVA]	Arm Capacitance C_arm_ (µF)	Arm Inductance L_arm_ [mH]	Arm Resistance R_arm_ [Ω]	Bus Filter Reactor [mH]
MMC1	±320	900	29.3	84.8	0.885	10
MMC2	±320	900	29.3	84.8	0.885	10
MMC3	±320	900	29.3	84.8	0.885	10
MMC4	±320	1200	39.0	63.6	0.67	10

**Table 3 sensors-22-09936-t003:** AC/dc System parameters.

dc System	Link12	Link13	Link34	Link24
Length [km]	100	200	100	150
Inductance [mH]	100	100	100	100
ac system	AC 1	AC 2	AC 3	AC 4
Rated voltage [kV]	400	400	400	400
Reactance X_ac_ [Ω]	17.7	17.7	17.7	13.4
Resistance R_ac_ [Ω]	1.77	1.77	1.77	1.34
Transformer µ_k_ [pu]	0.15	0.15	0.15	0.15

**Table 4 sensors-22-09936-t004:** Cable parameters.

Cable	Outer Radius [mm]	[Ωm]	Є_re1_ [-]	µ_re1_ [-]	Link34
Core	19.5	1.7 × 10^−8^	--		1
Insulation	48.7	--	2.3	150	1
Sheath	51.7	2.2 × 10^−7^	--	100	1
Insulation	54.7	--	2.3	AC4	1
Armor	58.7	1.8 × 10^−7^	--	400	10
Insulation	63.7	--	2.3	13.4	1

**Table 5 sensors-22-09936-t005:** Internal fault scenarios for training data.

Transient Period	Training Samples	Fault Resistance (Ω)	Fault Distance (km)	Noise (dB)
10 ms	357	0.01, 25, 50, …, 375, 400	1, 10, 20, …, 180, 190, 198	20, 25, 30
Total faulty sample = 357/each fault type; dc-link faults are first classified into two parts: pole to pole and pole to ground fault. Therefore, total training samples = ***k*** = (**F_int_** = 357 ∗ 2) = 714. Fault distance is noted from MMC1 to MMC 3 and MMC1 to MMC2, respectively.				

**Table 6 sensors-22-09936-t006:** Optimized parameters.

Hyperparameters	Range	Fault Location Model
Learning Rate	[1 × 10^−2^–1]	0.010037
Hidden Layers/Neurons (NHL)	[1–40]	28
Momentum	[0.001–0.005]	0.0028608
Epochs	[20–1000]	994
Gradient	[1 × 10^−7^–10^−6^]	1.2925 × 10^−7^
Validation	[0–6]	4

**Table 7 sensors-22-09936-t007:** Testing fault scenarios for testing data.

Transient Period [10 ms]	Testing Samples	Fault Resistance (Ω)	Fault Distance (km)	Noise (dB)
10 ms	400	10, 35, 60, 85, …, 435, 460, 485	5, 15, 25, …, 175, 185, 195	20, 25, 45
Total faulty sample = 400/each fault type, Total testing samples = [(400) ∗ 2] = 800, Refer Table 6 for fault distance				

**Table 8 sensors-22-09936-t008:** Fault location estimation errors.

Fault Type	Total Faults	Max Absolute Error (km)	Max Percentage Error (%)	Overall Absolute Error (km)	Overall Percentage Error (%)
PTP	400	2.6350	1.3174	0.9853	0.4927
PTG	400	2.6412	1.3206	1.0723	0.5361
Average Error	NA	NA	NA	1.0288	0.5144

**Table 9 sensors-22-09936-t009:** Fault resistance estimation errors.

Fault Location	Fault Resistance (Ω)	Fault Type			dc-Link Fault Location Results					
					Predicted Location		Absolute Error		Percentage Error (%)	
					PTP	PTG	PTP	PTG	PTP	PTG
5 km of dc link	10	PTP	PTG		5.02021	5.03022	0.02021	0.03022	0.01011	0.01511
	110	PTP	PTG		5.07141	5.08142	0.07141	0.08142	0.03571	0.04071
	260	PTP	PTG		5.63413	5.76481	0.63413	0.76481	0.31707	0.38241
35 km of dc link	35	PTP	PTG		35.05123	35.07134	0.05123	0.07134	0.025615	0.03567
	235	PTP	PTG		35.62858	36.10184	0.62858	1.10184	0.31429	0.55092
	285	PTP	PTG		35.86144	36.31471	0.86144	1.31471	0.43072	0.65736
125 km of dc link	260	PTP	PTG		126.67141	126.81487	1.67141	1.81487	0.83571	0.90744
	385	PTP	PTG		126.76175	126.91231	1.76175	1.91231	0.88088	0.956155
	110	PTP	PTG		126.01522	126.52812	1.01522	1.52812	0.50761	0.76406
185 km of dc link	260	PTP	PTG		186.94571	186.75387	1.94571	1.75387	0.972855	0.87694
	385	PTP	PTG		183.63147	186.93141	1.36853	1.93141	0.68427	0.96571
	110	PTP	PTG		186.34578	186.53681	1.34578	1.53681	0.67289	0.76841
Normal operation	X	X	X	X	NOT APPLICABLE		NOT APPLICABLE		NOT APPLICABLE	

**Table 10 sensors-22-09936-t010:** Results under the different noisy event.

Noise (dB)	Fault Location	Fault Resistance (Ω)	Fault Type		dc-Link Fault Location Results					
					Predicted Location		Absolute Error		Percentage Error (%)	
					PTP	PTG	PTP	PTG	PTP	PTG
25	5 km of dc link	10	PTP	PTG	5.04512	5.06727	0.04512	0.06727	0.02256	0.03364
		110	PTP	PTG	6.01202	6.03567	1.01202	1.03567	0.50601	0.51784
		260	PTP	PTG	6.26783	6.15872	1.26783	1.15872	0.63392	0.57936
20	45 km of dc link	35	PTP	PTG	46.06982	46.23672	1.06982	1.23672	0.53491	0.61836
		235	PTP	PTG	46.84612	47.03452	1.84612	2.03452	0.92306	1.01726
		285	PTP	PTG	47.03487	46.76324	2.03487	1.76324	1.01744	0.88162
45	155 km of dc link	260	PTP	PTG	156.96342	154.06853	1.96342	0.93147	0.98171	0.46574
		385	PTP	PTG	154.13647	153.02356	0.86353	1.97644	0.43177	0.98822
		110	PTP	PTG	154.43628	154.36571	0.56372	0.63429	0.28186	0.31715

## Data Availability

Not applicable.

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
