# Peer review of "Intelligent Sensors for dc Fault Location Scheme Based on Optimized Intelligent Architecture for HVdc Systems"

_sensors, 2022, doi:10.3390/s22249936_

Round 1

Reviewer 1 Report

1.      This scientific paper aims to use the single-ended fault location technique using a multilayer feed-forward neural network to locate the DC-link faults during dynamic operations. Bayesian optimization (BO) based artificial neural network (ANN) is used to train characteristic features efficiently. Selected features are fed into a BO-based tuned multilayer ANN model in the evaluation stage. The work is nicely shown and explained in detail. However, the innovation of this paper should be rewritten through several points.

2.      Particularly innovation in the introduction section needs to be more precise.

3.      How generalizable is the approach proposed?

4.      It is best to compare your research with other research to better highlight your article’s contribution.

5.      The proposed control design will facilitate the multi-vendor realization of hybrid AC/DC protection systems?

6.      In figure 7, please add the SNR abbreviation. It is missing.

7.      The table header format is not uniform. The header of Tables 1-6 is capitalized, but the header of table 7 is not.

8.      The generated samples for each fault location up to 8 levels using DWT-MRA to obtain wavelet coefficients. Please explain why you selected level 8.

9.      One more important general note. It would help if you improved the formatting of the article. Last but not least, you must consider the format template of MDPI.

10.  The references‘ format is not unified. For example, the years of references 31 and 17 are missing information. It is suggested to carefully check the format of references and unify the format of references.

11.  Some of the references are very old, and it is better to add recent ones, for example, references 4 and 27.

12.  The overall conclusion is written well. It should not be regarded as a summary section but highlight the most striking results. However, future study is missing; please add that.

13.  Overall paper is good, and I would recommend it for further consideration.

Reviewer 2 Report

In this article, the authors present a method for optimizing the novel model to optimize the internal and external parameters of the neural network. Moreover, they have provided compact features to locate dc fault locations. The work is nicely shown and explained in detail. The topic is essential to be included in the energy grids, especially in cases of renewable energy farm protection. I have some comments:

1.      The focus and tasks of the paper need to be better defined and described.

2.      Why has the author chosen a 10 ms window? Can a variable time window be added to reduce the computational burden?

3.      Why have the authors chosen cable instead of overhead lines?

4.      The proposed model is selective or unselective?

5.      One more important general note. You must significantly improve the formatting of the article. This concerns the presentation of the equations and the writing of electrical quantities in the text.

6.      The introduction section's last paragraph must include the missing objectives.

7.      Kindly review for the presence of grammatical errors in the text.

8.      Highlight the conclusion with major findings and the contribution the paper proposes.

Reviewer 3 Report

This paper reports the intelligent scheme for dc fault location in the VSCs-based HVDC system connected to a common AC system. Generally, it is well organized and written, and I have some suggestions listed below:

1) Please clearly state and indicate the impact of renewable energy sources, such as wind power plants, on the presented problem in your manuscript.

2) The background information is recommended to be enriched from a broad framework. For instance, accurate fault location is also benefical for power system voltage stability [R1] and efficient electricity market operation [R2].

R1. Zhang M, Li J, Li Y, et al. Deep learning for short-term voltage stability assessment of power systems[J]. IEEE Access, 2021, 9: 29711-29718.

R2. Xiao D, Chen H, Wei C, et al. Statistical Measure for Risk-Seeking Stochastic Wind Power Offering Strategies in Electricity Markets. Journal of Modern Power Systems and Clean Energy, 2022, 10(5): 1437-1442.

3) Please address Section II in detail and be careful if find any mistakes because that section has a lot of mathematical equations.

4) Can the impact of wind uncertainty also be studied?

5) Why choose the VSCs connected to a common AC system as the research object?

6) Please check the text for any grammatical errors.

7) Highlight the conclusion with key findings and the paper's proposed contribution.

8) Bayesian optimization (BO) based artificial neural network (ANN) is used to train characteristic features efficiently. Why did ANN choose it, and do you think it is good for hardware implementation? Please state that.

9) The work is subtly showcased and clearly stated. However, the paper's innovation should be rewritten in several places.
